# Exploring the Pathogenesis and Mechanism-Targeted Treatments of Rosacea: Previous Understanding and Updates

**DOI:** 10.3390/biomedicines11082153

**Published:** 2023-07-31

**Authors:** Chengqian Chen, Peiru Wang, Linglin Zhang, Xiaojing Liu, Haiyan Zhang, Yajing Cao, Xiuli Wang, Qingyu Zeng

**Affiliations:** Institute of Photomedicine, Shanghai Skin Disease Hospital, School of Medicine, Tongji University, Shanghai 200040, China

**Keywords:** rosacea, inflammation, inflammatory skin disease, pathogenesis, therapeutic strategies, treatment options

## Abstract

Rosacea is a chronic inflammatory skin disease characterized by recurrent erythema, flushing, telangiectasia, papules, pustules, and phymatous changes in the central area of the face. Patients with this condition often experience a significant negative impact on their quality of life, self-esteem, and overall well-being. Despite its prevalence, the pathogenesis of rosacea is not yet fully understood. Recent research advances are reshaping our understanding of the underlying mechanisms of rosacea, and treatment options based on the pathophysiological perspective hold promise to improve patient outcomes and reduce incidence. In this comprehensive review, we investigate the pathogenesis of rosacea in depth, with a focus on emerging and novel mechanisms, and provide an up-to-date overview of therapeutic strategies that target the diverse pathogenic mechanisms of rosacea. Lastly, we discuss potential future research directions aimed at enhancing our understanding of the condition and developing effective treatments.

## 1. Introduction

Rosacea is a common chronic inflammatory cutaneous disorder that affects about 5.46% of the global adult population [1]. It primarily affects the central facial skin and presents with symptoms such as recurrent episodes of flushing, persistent erythema, telangiectasia, papules, pustules, edema, phymatous changes, or a combination of these symptoms. Rosacea can be classified into four subtypes based on these symptoms: erythematotelangiectatic rosacea (ETR), papulopustular rosacea (PPR), phymatous rosacea (PhR), and ocular rosacea (OR) [2,3]. Although the pathophysiological mechanisms of rosacea remain unclear, the prevailing consensus is that the condition primarily stems from immune dysregulation and/or neurovascular dysfunction, as well as an impaired skin barrier. Triggers such as ultraviolet radiation, temperature changes, diet, and stress can exacerbate the underlying innate immune response and/or neurovascular dysfunction [4]. Recent studies have also highlighted the role of microbial dysbiosis, neuroimmune interactions, metabolic dysfunction, and sebaceous gland dysregulation in the development of rosacea. Other factors such as genetic predisposition and oxidative stress are also thought to play a role (Figure 1).

Regarding treatment, various guidelines and expert consensus offer a range of therapeutic options tailored to different phenotypes [2,5,6,7]. In terms of addressing the pathogenesis of rosacea, the most traditional and commonly employed approach is through the use of anti-inflammatory treatments. Novel drugs targeting neurological and psychological factors have recently gained attention. Several other therapeutic options have emerged targeting other specific pathways, including vascular dysregulation, and microbial dysbiosis. New formulations or routes of administration for some drugs are also being explored. Physical therapies, such as laser and photodynamic therapy, have also shown promising results in managing the symptoms of rosacea. Combination therapy with multiple agents has also shown potential for synergistic effects and improved clinical outcomes in the treatment of rosacea.

In this article, we provide a detailed and comprehensive investigation of the pathogenesis of rosacea, with a particular focus on the emerging and novel mechanisms that may contribute to its development. Drawing from both the current and emerging research, our review offers a thorough and up-to-date overview of the various therapeutic strategies that target the pathogenic mechanisms underlying rosacea. Finally, we present an outlook on potential future research directions that aim to enhance our understanding of the pathogenesis of rosacea and advance the development of novel and effective treatment approaches for this prevalent dermatological disorder.

## 2. Pathogenesis

### 2.1. Immune Dysregulation

Rosacea is characterized by immune dysfunction involving both innate and adaptive immunity, with alterations to the TLR2/KLK5/LL37 pathway being the most well-studied pathological mechanism (Figure 2). Toll-like receptors (TLRs) are a family of transmembrane receptors expressed on various skin cell types, including keratinocytes (KCs) and fibroblasts, which can recognize pathogen-associated molecular patterns (PAMPs) and damage-associated molecular patterns (DAMPs). Additionally, TLRs are also expressed by various resident innate immune system cells in the skin, including dermal mast cells (MCs), phagocytes, and dendritic cells, as well as epidermal Langerhans cells, and recruitable cells from the blood, such as neutrophils, macrophages, and γ-δ T cells [8,9,10]. TLR stimulation activates innate immunity, leading to the controlled activation of nuclear factor-kappa B (NF-κB) signaling and the subsequent production of cytokines, chemokines, and cathelicidin for host defense [11,12,13]. However, excessive TLR activation can result in inappropriate inflammation. In rosacea patients, TLR2 is overexpressed in KCs and infiltrating dermal cells at the site of skin lesions [11], leading to an overactive innate immune response. SIRT7 downregulation in aging skin cells has been shown to downregulate TLR2 and inhibit NF-κB pathway activation, potentially explaining the decreased incidence or symptom relief of rosacea in the elderly population [14]. Increased tumor necrosis factor (TNF)-α or LL37 signaling in rosacea patients can directly activate the transcription factor NF-κB, inducing the expression of interleukin (IL)-1α, IL-1β, IL-18, and IL-33 [15]. The arylhydrocarbon receptor (AhR), when activated, can improve rosacea-like skin lesions and inhibit the expression of TLR2 and downstream chemokines, such as chemokine (C-C motif) ligand (CCL) 5 [16]. Following TLR2 stimulation, AhR has been found to promote the expression of inflammatory genes by increasing JNK/mitogen-activated protein kinase signaling and FosB expression, while inhibiting the expression of TLR2 to prevent excessive amplification of inflammation [17].

As an important molecule for host defense against pathogenic microorganisms, cathelicidin is an antimicrobial peptide critical to innate immunity [18]. In humans, the only endogenous cathelicidin is LL37/hCAP-18, which is expressed in both epithelial cells and leukocytes [19,20,21,22]. Trypsin-like serine protease kallikrein 5 (KLK5) cleaves hCAP18 encoded by the human Camp gene into its active form, LL37. KLK5 has been shown to be upregulated in patients with rosacea, leading to the aberrant accumulation of active LL37 [15]. Although LL37 is primarily responsible for its antibacterial function, it can also promote inflammation through multiple pathways. Specifically, LL37 activates the janus kinase (JAK)/signal transducer and activator of transcription (STAT) pathway, inducing the upregulation of pro-inflammatory cytokines, such as TNF-α, IL-6, IL-8, and monocyte chemoattractant protein-1 [15]. Furthermore, LL37 binds to TLR2 on KCs to activate the mammalian target of rapamycin complex 1 (mTORC1) signaling pathway, leading to increased cathelicidin expression and the formation of a positive feedback loop that amplifies and sustains inflammation. LL37 also activates the NF-κB signaling pathway through the mTORC1 signaling pathway [23]. Additionally, LL37 promotes leukocyte chemotaxis and mast cell degranulation, inducing the release of pro-inflammatory mediators, such as IL-6 and matrix metalloproteinase 9 (MMP9). Neutrophils recruited during this process express LL37 and MMP9 [15]. MMP-9 activates KLK5 by cleaving its preproenzyme form [24], leading to the cleavage of hCAP18 into LL-37 peptide and augmenting the inflammatory response. In addition, LL37 promotes the assembly and activation of the NOD-like receptor family pyrin domain containing 3 (NLRP3) inflammasome, inducing cell death and activating many other pro-inflammatory factors, such as IL-8, TNF-α, and MMPs, to maintain innate immunity [25]. Thus, the intradermal injection of LL37 has become a widely used animal model for studying rosacea [26]. Additionally, two novel antimicrobial peptides, S100A15 and S100A9, which are elevated in skin lesions of rosacea and contribute to chronic inflammation, have gained attention [27,28,29,30]. NEAT1, a long non-coding RNA, has been shown to promote inflammation by upregulating the expression of S100A9 [31].

Mast cells represent one of the main sources of cathelicidin and its activating enzymes in the skin [32] and are markedly increased in various subtypes of rosacea lesions [33,34,35]. The involvement of MCs in LL37-induced rosacea inflammation in mice has been confirmed by the demonstrated alleviation of these effects through the inhibition of MC activation or degranulation or direct MC ablation [36,37,38]. The classical pathway of MC activation is IgE-mediated degranulation but can also be mediated by other pattern recognition receptors (PRRs), such as NLRP3 and TLRs [4]. LL37 accumulated abnormally within rosacea skin lesions upregulates the expression of multiple PRRs, such as TLRs [39,40], on mast cells, which can enhance their ability to detect invading pathogens. LL37, in conjunction with neuropeptides (NPs), another key mediator involved in the pathogenesis of rosacea, can activate MCs through the Mas-related G protein-coupled receptor member X2 (MRGPRX2, mouse homologue MrgprB2) [41,42,43]. Once activated, MCs can release histamine, tryptase, and chymase stored in granules, along with various cytokines such as IL-1, TGF-β, TNF-α, and vascular endothelial growth factor (VEGF) [4], to further activate MCs themselves. Additionally, activated MCs express MMP9 to promote the production of LL37 [38]. In conclusion, LL37-mediated MC activation leads to further production of LL37 by activated MCs, resulting in amplified inflammation. Furthermore, MCs recruit fibroblasts and promote their proliferation to contribute to rosacea fibrosis [38,44].

Neutrophil and macrophage infiltration are prominent features of rosacea lesions [45,46]. Neutrophils contribute to a positive feedback loop by expressing nitric oxide (NO), reactive oxygen species (ROS), LL37, and MMP9, promoting inflammation at the sites of lesions [15]. Interestingly, some N2 neutrophils in rosacea patients and mouse models exhibit anti-inflammatory effects by regulating vascular factors and inhibiting CD4+ T cell proliferation [47]. Macrophages also play a role in rosacea pathogenesis by expressing TLR2 and NLRP3, which activate inflammation [48,49,50]. ADAM-like Decysin-1 (ADAMDEC1) and guanylate-binding protein 5 (GBP5) are particularly highly expressed in rosacea tissues and participate in the polarization of M1 macrophages, resulting in the release of a broad spectrum of pro-inflammatory cytokines and chemokines [48,51].

The current understanding of adaptive immunity in rosacea is limited, with studies indicating a high prevalence of CD4+ T cell infiltration around hair follicles and blood vessels in affected skin lesions, while there is no significant increase in CD8+ T cells [46,52,53,54]. These infiltrating T cells tend to polarize towards Th1 and Th17, expressing Th1-signature cytokines IFN-γ and TNF-α, as well as Th17-signature cytokine IL-17 [46,55]. Elevated IL-17 levels in rosacea patients have been linked to angiogenesis, inflammation, and the induction of MMP-9 and LL37 expression [56,57]. *Demodex* mite infestation is associated with increased levels of Treg cells and Th9 cells [58]. As for humoral immunity, while CD20+ B cells comprise 10–20% of the infiltrating inflammatory cells [33,46], their precise role needs further research.

Overall, while the involvement of innate immune cells in the pathogenesis of rosacea has been extensively investigated, the roles of adaptive immunity and specialized cells in this condition are still largely unknown (Figure 3). Given the important functions of these cells in immune responses, further studies are necessary to better understand their contributions to the development of rosacea. For instance, sebocytes, known for sebum secretion, produce immunomodulatory molecules relevant to inflammatory skin conditions, but their precise roles in rosacea inflammation are unclear. Interactions among immune cells, the nervous system, blood vessels, and microbiota should also be considered for a comprehensive understanding of rosacea development.

### 2.2. Vascular, Neurovascular, and Neuroimmune Dysregulation

Rosacea is a multifaceted skin disorder characterized by facial erythema and flushing, which can be attributed to various physiological changes, including increased skin blood flow, vasodilation, angiogenesis, elevated permeability, and upregulated levels of vascular endothelial growth factor (VEGF), a critical mediator of angiogenesis and vasopermeability [59,60,61,62,63]. However, conventional drugs that focus on constricting blood vessels have limited efficacy and duration. This suggests that vascular dysregulation alone does not fully explain blood vessel regulation in rosacea. Recent research suggests the involvement of the nervous system and intricate neuroimmune interactions. (Figure 4). These findings present opportunities for innovative therapeutic strategies targeting neurovascular or neuroimmune communication in rosacea.

The complex pathophysiology of vascular dysregulation involves the activation of multiple pro-angiogenic mediators. One such mediator, LL37, activates the formyl peptide receptor 1 (FPR1) receptors, epidermal growth factor receptors (EGFR), and downstream signaling in epithelial cells to promote angiogenesis. LL37 also increases VEGF levels in epidermal KCs and the expression of adhesion molecules in endothelial cells [15]. Moreover, LL37 induces angiogenesis in human endothelial cells through mTORC1 signaling [64]. MCs, highly activated in rosacea skin lesions, release a multitude of pro-angiogenic molecules, such as vascular endothelial growth factor (VEGF), fibroblast growth factor (FGF), histamine, and tryptase [4]. Lee et al.’s study found that upregulated Hippo signaling in rosacea skin lesions also participates in VEGF-mediated angiogenesis [65]. Systemic metabolic changes may also play a role, as elevated levels of glutamic acid and aspartic acid in rosacea patients can stimulate nitric oxide production, contributing to vessel dilation [66]. This novel insight warrants further investigation to fully elucidate.

Patients with rosacea present with sensitivity to various triggering factors, such as cold, heat, ultraviolet (UV) radiation, capsaicin, alcohol, and stress, which are defining characteristics of sensitive skin. Neurological and psychiatric factors have also been implicated in the onset of rosacea, as evidenced by their association with dementia, sleep disorders, depression, and anxiety [67,68,69]. Transient receptor potential (TRP) ion channels are expressed in sensory neurons and non-neuronal cells. These channels mediate various symptoms triggered by different stimuli, such as flushing and sensitivity [70]. Elevated levels of TRP channels and neuropeptides have been documented in both neuronal and non-neuronal cells across all subtypes of rosacea [71,72,73,74]. As a result, these cells are more prone to activation by external specific stimuli. Upon activation, various vasodilatory neuropeptides, such as pituitary adenylate cyclase-activating polypeptide (PACAP), substance P (SP), vasoactive intestinal peptide (VIP), and calcitonin gene-related peptide (CGRP), are released, which mediate a range of effects, including pain, itching, vasodilation, and increased permeability [15,70]. An abnormal release of serotonin (5-HT) from MCs and platelets, as well as its involvement in thermoregulation, may also play a role in the facial vascular dysfunction observed in rosacea [75,76,77].

Recent studies have shed light on the intricate interplay between the nervous and immune systems. Upon the activation of TPR ion channels, the release of CGRP and SP by neurons binds to receptors on endothelial cells, fibroblasts, and some immune cells, inducing the production of other pro-inflammatory mediators, and thereby contributing to the maintenance of cutaneous neurogenic inflammation in rosacea [78]. Immune cells in the skin barrier are often found in close proximity to peripheral nervous system nerve fibers. These immune cells express receptors for neuropeptides and neurotransmitters, which enable them to be modulated by the nervous system [79,80]. Meanwhile, neurons have been observed to express immune-related receptors, like PRRs and cytokine receptors [81,82,83], to mediate neurogenic inflammation and receive immune cell regulation. MCs are increasingly recognized as important contributors to neuroimmune crosstalk, particularly in the skin [78]. Evidence suggests that MCs and sensory neurons form closely associated clusters in the skin, allowing for rapid communication between the two cell types [84]. The activation of sensory neurons results in the release of NPs, which can act on MRGPRX2/MrgprB2 receptors and then prompt MC degranulation, as well as chemokine/cytokine production, thereby mediating inflammation and vasodilation [38,71,84,85]. Conversely, histamine, tryptase, and other mediators released by MCs have been shown to sensitize sensory neurons, generating action potentials that mediate pain and itching, and inducing the release of NPs in peripheral endings [81,86]. This creates a bidirectional loop between MCs and sensory neurons, leading to neurogenic inflammation.

Neurological aspects of rosacea pose significant clinical challenges due to recurring and resistant episodes. The intricate interplay among the nervous system, vasculature, and inflammation necessitates thorough exploration to comprehend underlying mechanisms. Despite several hypotheses, the precise pathogenesis of neurological symptoms in rosacea remains elusive. Elucidating the mechanisms underlying the association between neurological dysfunction and rosacea may help identify novel therapeutic targets for this condition.

### 2.3. Skin Barrier Dysfunction

The integrity of the skin barrier is of utmost importance in shielding the body against external agents and preserving its internal balance. The main components responsible for the epidermal permeability barrier include the stratum corneum and tight junctions [87]. Patients with rosacea display dry and sensitive facial skin, which is associated with a notable elevation in the pH value, increased transepidermal water loss (TEWL), and a marked reduction in stratum corneum hydration [88,89,90,91]. These changes are primarily attributed to the decreased expression of claudins (CLDNs) [92], which represent the main components of tight junctions. In a study conducted by Medgyesi et al., molecular-level alterations in key components of the skin barrier, including CLDNs, LOR, and KRT1, were observed in PPR [93]. Proper regulation of the calcium ion concentration gradient within the epidermis is vital in maintaining skin barrier formation, permeability barrier homeostasis, and keratinocyte differentiation. Disruptions in the intracellular and extracellular calcium ion concentrations due to the abnormal expression and function of TRP channels in rosacea patients could impair skin barrier function [94,95]. Skin barrier dysfunction can contribute to the development and progression of inflammatory skin disorders. Specifically, an impaired skin barrier can trigger the activation of STAT3 in keratinocytes, prompting the release of inflammatory cytokines that activate STAT1 in immune cells, thereby prompting the release of additional inflammatory cytokines that can further disrupt the skin barrier (Figure 5) [96]. The section below expounds on changes in lipids and microbiota, which represent critical components of the skin barrier as well. These elements operate in conjunction with the conventional physical barrier of the skin. For instance, a study has indicated that the knockout of Claudin-1 could lead to sebaceous gland damage and impede the functionality of sebaceous gland holocrine secretion [97]. Furthermore, the breakdown of the skin barrier can also promote alterations in the composition of microbiota, which may potentially participate in the pathogenesis of rosacea. Hence, it is evident that the disruption of the skin barrier plays a crucial role in the complex network of pathogenesis in rosacea. Recent attention has been increasingly focused on the functionality of the skin barrier.

### 2.4. Microbial Dysbiosis

The cutaneous microbiota, consisting of a diverse array of microorganisms, such as bacteria, viruses, fungi, and mites that colonize both on and within the skin, constitutes a crucial constituent of the skin’s barrier function. The composition of the microbiota is shaped by various factors, such as gender and age, as well as local environmental factors, such as pH, temperature, and lipid composition. Notably, changes in the facial skin microbiota have been observed in rosacea patients using 16S rRNA gene sequencing [98]. Among the skin microbiota linked to the inflammatory response in rosacea, *Demodex* mites (and their associated bacterium, *Bacillus oleronius*) stand out (Figure 5). However, the causative relationship between the mites and rosacea remains contentious, with current evidence indicating a bidirectional causality between the two. *Demodex* mites, specifically *Demodex folliculorum* and *Demodex brevis*, inhabit hair follicles and meibomian/sebaceous glands, respectively [99]. While their density is found to be higher in rosacea patients and is positively correlated with disease severity, it is not necessarily a factor in the development of the disease [15,100,101,102,103]. Hair follicles are immune-privileged and can tolerate moderate colonization of *Demodex* mites, allowing for a harmonious symbiosis. Mites have developed mechanisms to evade immune responses and establish a persistent presence in the follicle [104]. Type 2 innate immunity is implicated in maintaining the symbiosis, and its diminished expression in PhR patients may lead to abnormal mite proliferation and subsequent hair follicle inflammation [105]. Skin barrier dysfunction and an imbalanced microbiota in rosacea patients can also lead to an escalation in the population of *Demodex* mites [106].

In cases where the density of *Demodex* mites is elevated, their allergens have been shown to activate the NLRP3 inflammasome and contribute to the activation of IL-1β [107]. Furthermore, chitin, present in the exoskeleton of mites, has been shown to induce an upregulation of TLR2 expression and a subsequent activation of TLR2, leading to the initiation of an immune response. Mites’ claws and mouthparts can directly damage tissues [108], while excessive mites obstruct hair follicles and sebaceous glands, disrupting the skin barrier [15,104,109]. Moreover, resident bacteria, enzymes, and feces from deceased mites may potentially exacerbate the inflammatory response [104,109]. The abnormal proliferation of *Demodex* can interfere with the integrity and repair of the skin barrier through the AhR signal [110]. Other microorganisms, including *Staphylococcus epidermidis*, have also been implicated in the induction of skin inflammation through TLR2 activation [24]. Overall, the pathogenesis of rosacea is multifactorial, involving an imbalance of several microorganisms, which, in turn, is affected by changes in the skin microenvironment. These factors collectively contribute to a vicious cycle that disrupts the balance of microbiota and exacerbates the disease.

Rosacea has been associated with gastrointestinal diseases such as *Helicobacter pylori* infection, inflammatory bowel disease, and small intestinal bacterial overgrowth (SIBO) [111,112,113,114,115]. Studies using 16S rRNA sequencing have demonstrated alterations in the microbial richness and composition of the feces of rosacea patients [116,117]. These results suggest that there may be a potential link between gut microbiota and the development of rosacea. Gut dysbiosis can impact the host’s immune system in complex ways, which can consequently affect the immune response of the skin [118,119]. Although the evidence is not conclusive and the composition of gut microbiota can differ significantly among individuals, some therapies for gastrointestinal diseases and microbiota have demonstrated effectiveness in ameliorating rosacea symptoms [118,120,121,122]. Moreover, a small yet significant difference in the blood microbiomes of rosacea patients compared to those of controls has been observed [123], though its clinical significance remains under investigation.

Overall, *Demodex* mites remain a research focus among the pathogenic microorganisms of rosacea. However, studies have been impeded by the inability to culture them in vitro. To overcome this obstacle, some studies have used chitin as a surrogate to evaluate the pro-inflammatory effects of *Demodex* mites on target cells. Nevertheless, recent research has revealed that the varying quantities of *Demodex* mites have distinct impacts on inflammation regulation [108], indicating that chitin may not be a sufficiently reliable substitute. With the continuous advancements in isolation and in vitro culture techniques of *Demodex* mites, we have a promising opportunity to further unravel their regulatory roles.

### 2.5. Metabolic Dysfunction

Rosacea has been associated with a number of metabolic-related disorders, including hypertension, dyslipidemia, thyroid disorders, obesity, and diabetes [124,125,126,127,128,129]. These observations suggest that systemic metabolic alterations may have a role in the pathogenesis of rosacea (Figure 5). In a study by Li et al., serum metabolic profiling revealed significant alterations in amino acids, fatty acids, organic acids, and carbohydrates in rosacea patients. The study also found increased levels of glutamic acid and aspartic acid, which promote the production of NPs and NO, leading to erythema and capillary dilation [66]. Another study showed that rosacea patients have distinct levels of bile acids, including elevated lithocholic acid, which can stimulate the production of inflammatory cytokines and chemokines via activation of the G protein-coupled bile acid receptor 1 (GPBAR1) [130]. Hypercholesterolemia has also been implicated in inflammation, as it leads to cholesterol accumulation in various immune cells, amplifying TLR signaling. Oxidized low-density lipoprotein can directly activate macrophages as ligands for TLRs, triggering pro-inflammatory signal transduction. Cholesterol can also induce NLRP3 inflammasome activation, further exacerbating inflammation [131]. Additionally, lower levels of serum bilirubin and uric acid, which possess antioxidant properties, were observed in rosacea patients [132]. The link between rosacea and diabetes may be attributed to the high levels of oxidative stress, which may lead to insulin resistance [133,134,135]. Previous studies have suggested an association between higher *Demodex folliculorum* density or infection rates and hyperglycemia caused by metabolic syndrome, polycystic ovary syndrome, gestational diabetes, and type 2 diabetes [136,137,138,139], but the underlying mechanism remains unclear. Skin barrier damage is commonly observed in obese patients, as indicated by reduced skin hydration [140]. However, studies have reported conflicting evidence regarding changes in TEWL in these patients [141,142]. Additionally, obese patients often exhibit disrupted sebaceous gland function due to elevated levels of androgens, insulin, growth hormone, and insulin-like growth factor [143,144].

Taken together, these findings highlight a complex relationship between rosacea and metabolic dysfunction, suggesting potential implications for the management and treatment of rosacea. Several dietary supplements are currently being evaluated as possible treatments for this condition. Further research is needed to elucidate the underlying mechanisms and explore potential therapeutic targets.

### 2.6. Sebaceous Gland Dysfunction

Rosacea lesions typically appear in the central facial region, where sebaceous glands are abundant. A specific subtype of rosacea, PhR, is associated with thickening of the skin due to sebaceous gland hyperplasia. These clinical findings suggest that sebaceous gland dysfunction may play a role in the pathogenesis of rosacea (Figure 5). This notion is supported by studies using topical isotretinoin, which demonstrated a significant reduction in sebaceous gland volume and sebum production, as well as improvement in erythema and papulopustules in patients with rosacea [143,145].

Sebum secretion by sebocytes is a crucial process that maintains skin hydration and reduces TEWL [146,147,148]. In rosacea, sebaceous gland dysfunction can arise from TLR-mediated inflammatory status, changes in the microbiota that colonize pilosebaceous unit [149,150], or dysregulated neural and endocrine influences [151,152]. These changes can lead to an altered sebaceous fatty acid profile, with increased levels of myristic acid and decreased levels of long-chain saturated fatty acids [106], rather than changes in the total amount of sebaceous fatty acids. These alterations in sebum composition can result in symptoms and signs of damaged skin barrier [153]. Sebum also serves as a substrate for the growth of some microorganisms. Perturbations in sebum composition can alter the composition of the skin microbiota, with potentially deleterious effects on skin health. Indeed, a retrospective study has shown a positive correlation between the size and density of sebaceous glands and the proliferation of *Demodex* mites in rosacea patients [154]. Furthermore, *Demodex* mites modulate TLR signaling in sebocytes, inducing the release of pro-inflammatory cytokines, such as IL-8, when their numbers reach a critical level [108]. Changes in sebum production may result in the downregulation of thymic stromal lymphopoietin (TSLP) expression in keratinocytes and sebocytes, thereby activating dendritic cells and T cell differentiation and promoting the infiltration of inflammatory cells [155]. Loss of TSLP expression results in RORγt+ innate lymphoid cells loss, leading to sebaceous gland hyperplasia and altered microbial symbiosis [156].

The sebaceous gland serves a vital role in maintaining skin immunity by producing a range of immune-regulating molecules, including lipids, cytokines, chemokines, and antimicrobial peptides [147,157,158]. Stimuli originating from both inside and outside the pilosebaceous unit, which have the ability to activate the TLR2 and TLR4 pathways, may trigger a rapid induction of an immune-competent state in sebocytes, resulting in the production of numerous cytokines, such as C-X-C motif chemokine ligand (CXCL)-8, CXCL-10, IL-1β, IL-6, CCL-5, and leptin [150,157,159]. Inflammatory cell infiltration around hair follicles, including mast cells, is a notable histological feature of rosacea and may be attributed to sebocyte chemotaxis [54,157,160,161,162]. However, the specific mechanism responsible for this characteristic perifollicular inflammation and the particular cells recruited, including those beyond mast cells, remain subjects that require further investigation.

According to current understanding, acne is widely considered as the skin disease most closely related to sebaceous gland dysfunction. It is clinically reasonable to see acne and rosacea occurring simultaneously, given the significant overlap in inflammatory factors. Although isotretinoin shows promise for treating both conditions by regulating sebaceous glands, the complexity of rosacea extends beyond this aspect. Therefore, it is important to consider a holistic approach to treating rosacea and to address the underlying causes of this condition beyond just sebaceous gland dysfunction.

### 2.7. Miscellaneous

Genetic Predisposition: Genetic predisposition is thought to play a role in the development of rosacea, as indicated by its family inheritance, twin concordance, and varying prevalence among different ethnic groups [163,164,165]. The current research has identified several genetic factors that may be associated with the development of rosacea. These factors include polymorphisms in genes, such as glutathione S-transferase (GST) [166], human leukocyte antigen (HLA) class II [167], tachykinin 3 receptor (TACR3) [168], vitamin D receptor (VDR) [169,170], and VEGF [171]. Additionally, a gain-of-function variant in the STAT1 gene has also been found to be implicated [172,173,174]. Recently, a notable study conducted whole-genome sequencing and whole-exome sequencing on samples from individuals within Chinese rosacea families. In this investigation, variant genes linked to neural function, including LRRC4, SH3PXD2A, and SLC26A8, were identified. Furthermore, follow-up experiments utilizing animal and cell models corroborated that mutations in LRRC4, among others, can facilitate neurogenic inflammation in rosacea by triggering the peripheral nerve-mediated secretion of NPs [175]. This research represents a significant advancement in understanding the genetic underpinnings of rosacea and its association with neural mechanisms, thereby offering valuable insights into the condition’s pathogenesis.

Cellular Stress Response: Oxidative stress dysfunction is linked to various inflammatory skin diseases [176,177]. Patients with rosacea exhibit elevated levels of oxidized disulfides in their serum, indicating an imbalance in oxidative stress [135,178]. Multiple mechanisms have been implicated in the development of rosacea through oxidative stress, including the production of ROS by neutrophils, lipid and protein peroxidation, and the promotion of an inflammatory state [178]. In individuals with rosacea, TNFα induction leads to the upregulation of Nav1.8, a sodium ion channel in epidermal keratinocytes. Nav1.8 binds to superoxide dismutase 2, inhibiting its antioxidant function and causing ROS accumulation, triggering pro-inflammatory signaling [179]. Additionally, endoplasmic reticulum (ER) stress plays a role in rosacea pathogenesis by stimulating cathelicidin production and increasing TLR2 expression [83,180]. However, it can also activate TLR2 signaling in neurons, leading to neurogenic inflammation [181].

Environmental Triggers: Numerous environmental factors have been identified as triggers or exacerbating agents of rosacea, with various underlying mechanisms involved. The activation of TRP ion channels and ER stress are among the pathways implicated in this condition, as previously described. UV radiation induces various skin responses, including ROS production [15], pro-inflammatory cytokine release (IL-33, IL-1β) [182,183], MMP upregulation [32], and angiogenesis stimulation (VEGF, FGF, IL-8 upregulation, thrombospondin-1 downregulation) [184]. Finally, it directly causes vasodilation through a thermal effect.

Alcohol, spicy food, cinnamaldehyde-containing foods, hot drinks, and histamine-rich foods are also known triggers or exacerbating factors of rosacea [185]. Alcohol causes vasodilation, inflammation, and oxidative stress, and long-term intake can lead to vasoregulation loss and gut microbiome dysbiosis [186,187,188]. Analogous to foods rich in histamine, the metabolism products of ethanol can cause the release of histamine and exacerbate symptoms [187,188]. Emotional stress activates the hypothalamic–pituitary–adrenal (HPA) axis, leading to cortisol release and activation of the inflammatory pathway, which impairs skin barrier function [189]. This can lead to further psychological burden on patients.

## 3. Treatments

In this chapter, we provide an overview of therapeutic strategies targeting various pathogenic mechanisms underlying rosacea (Table 1). While well-established treatment methods are briefly reviewed, our main focus is on emerging therapies. It should be noted that although a certain drug is classified under a certain mechanism, it may also work through other mechanisms.

### 3.1. Anti-Inflammatory Strategies

Azelaic acid gel (15%) is a classic and clinically established medication that has gained US Food and Drug Administration (FDA) approval for mild-to-moderate rosacea treatment by suppressing KLK5 and cathelicidin expression, activating PPARγ, and reducing pro-inflammatory factors [190,191]. Another inhibitor of trypsin-like serine protease, ε-aminocaproic acid, has recently demonstrated a beneficial impact on the severity of rosacea in a small, randomized pilot trial [192].

Tetracycline antibiotics are widely used to treat rosacea due to their anti-inflammatory effects. However, unlike acute infections, rosacea treatment lasts for weeks to months, and therefore, sub-antibiotic doses (40 mg) of doxycycline, which have been approved by the FDA, are often preferred to avoid bacterial resistance or dysbiosis. Sub-antibiotic doses of doxycycline effectively reduce erythema and inflammation by inhibiting chemotaxis, ROS production, and MMPs [2,7]. Safer topical formulations (such as minocycline foam) and a new generation of tetracyclines, sarecycline, are gaining attention as potential treatments for rosacea [193,194,195,196].

Oral isotretinoin is a recommended option for granulomatous rosacea, early soft phymatous changes, and refractory erythema and papulopustules, albeit with a need for caution due to its potential teratogenic effects [2,5,7,197]. The efficacy of isotretinoin is thought to be linked to its ability to regulate innate immunity by negatively modulating the expression of TLR2 in keratinocytes [198]. Moreover, isotretinoin is known to reduce sebum production and sebaceous gland size, thus improving disrupted sebaceous gland function [143,145,199].

In accordance with Swiss guidelines, pimecrolimus 1%, a calcineurin inhibitor, is a recommended treatment for erythema and papulopustular lesions [2]. Pimecrolimus 1% inhibits T cell and mast cell activation [200], but caution is needed due to potential rosacea-like eruptions, which may be due to its immunosuppressive effect resulting in the overgrowth of microorganisms such as *Demodex folliculorum* [2,201]. Similarly, tacrolimus also exhibits this double-edged sword effect [201].

Hydroxychloroquine, an anti-malaria drug with anti-inflammatory properties, is used to treat systemic autoimmune diseases. In rosacea, it reduces skin inflammation by inhibiting mast cell activation caused by LL37 and calcium influx [37]. A randomized trial compared hydroxychloroquine to doxycycline and found similar efficacy and safety [202]. Artemisinin, another anti-malarial drug, and its bioactive derivative, artesunate, suppress the expression of inflammatory biomarkers induced by LL37 via the inhibition of various transcription factors, including NF-κB, mTOR, and STAT [203,204,205]. A randomized pilot study including 130 subjects found that artemether, a lipid-based derivative of artemisinin, showed higher effectiveness and lower papule and pustule scores than metronidazole emulsion after 4 weeks of treatment, although there was no significant difference in the erythema score [206].

Tranexamic acid is a plasmin inhibitor used for treating bleeding conditions. In dermatology, it is used off-label for melasma and shows potential for rosacea treatment. Studies have found that tranexamic acid reduces the mast cell count in the skin [207]. It also suppresses inflammatory biomarkers and angiogenesis [208]. Kim et al. reported that soaking with tranexamic acid solution once or twice a week improved erythema and discomfort symptoms in six rosacea patients [209], and in an open-label trial with a small sample size, local application of tranexamic acid solution was found to improve symptoms in patients with ETR [210]. Daadaa et al. reviewed six ETR patients who received intradermal microinjections of tranexamic acid, and the Investigator Global Assessment (IGA) of Rosacea Severity Score decreased by an average of 2.4 ± 0.5 [211]. Moreover, tranexamic acid can promote skin barrier repair by inhibiting serine protease and a physical interaction between the urokinase-type plasminogen activator and the stratum corneum [212,213]. A randomized, vehicle-controlled, split-face study involving 30 patients showed that local tranexamic acid treatment significantly decreased the TEWL and skin surface pH value of rosacea patients, enhanced hydration of the stratum corneum, and reduced the number of inflammatory lesions [214].

ACU-D1, a novel 26S proteasome inhibitor, has been found to exhibit anti-inflammatory effects by inhibiting the activation of NF-κB. In a double-blind, randomized, placebo-controlled study involving 40 patients with moderate to severe rosacea, ACU-D1 was shown to be effective in reducing inflammatory lesions in 92% of patients, with 27% experiencing a 2 plus grade IGA reduction of clear to nearly clear [215].

IL-17 is produced primarily by Th17 cells and promotes inflammation in multiple ways. IL-17 blockade has been successfully used in treating psoriasis and psoriatic arthritis, and may also be effective in treating rosacea [216]. However, the cost of treatment can be high. In an open-label, rater-blinded, investigator-initiated study, secukinumab treatment (300 mg weekly for 5 weeks followed by monthly dosing for 2 months) significantly reduced the number of papules and the global severity score in 17 patients [217]. Nonetheless, the risk of infections should be taken into account. Further high-quality randomized controlled studies are needed to draw definitive conclusions.

### 3.2. Vascular-Targeted Strategies

Brimonidine tartrate gel acts as a potent vasoconstrictor by binding to α2-adrenergic receptors on the smooth muscle surrounding the facial skin blood vessels [218]. Due to its quick onset of action and safety profile, it is widely used to treat transient and persistent facial erythema rather than papules and pustules [2,5,6,7,197]. In a 12-month open-label observational study, topical brimonidine was found to be both safe and effective for maintenance therapy in patients with rosacea [219]. Another topical α1 agonist, oxymetazoline, has also been approved for the treatment of persistent erythema in the United States [5,197].

Timolol, a non-selective β-adrenergic receptor blocker, has vasoconstrictive and anti-angiogenic effects. It inhibits inflammatory mediators, like MMPs and IL-6, and downregulates VEGF to hinder angiogenesis [220,221]. When used topically, timolol also promotes keratinocyte migration and skin barrier repair [222]. To investigate the potential therapeutic effects of timolol on rosacea, a pilot clinical trial involving eight patients found that long-term (12 weeks) topical application of timolol 0.5% gel-forming solution significantly improved erythema, although rebound occurred after discontinuation [223]. In another study, 16 patients with mild to moderate ETR were enrolled in a randomized, single-blind, placebo-controlled split-face study, where the side of the face treated with timolol maleate 0.5% eye drops applied with wet compresses every night for 28 days showed significant improvements in the Clinician Erythema Assessment (CEA) and a patient self-assessment (PSA). Although one case of local adverse reaction occurred, it resolved on its own [224]. However, a single-arm clinical study involving 58 patients with ETR and PPR found that although topical timolol 0.5% improved clinical parameters of rosacea, it did not reach statistical significance. The mean percentage of improvement in telangiectasia and erythema was higher (50% and 41.38%, respectively), while the improvement in papules and pustules was only 7.41% [220].

### 3.3. Targeting Neurological and Psychological Factors

Recent interest in botulinum toxin type A as a potential treatment for rosacea has grown due to its ability to inhibit vasodilating acetylcholine and regulate neuropeptides [225,226], reduce the mast cell count, suppress mast cell degranulation, and decrease the expression of certain MMPs in skin fibroblasts [227,228]. Despite some uncertainty about its mechanism of action, a number of studies have suggested that botulinum toxin may also be capable of reducing sebum production and increasing skin hydration [228,229]. A 2021 systematic review analyzed nine studies with a total of 130 participants, reporting satisfactory efficacy and safety, but limited by small sample sizes, imperfect study designs, and short follow-up times [230]. Another randomized, controlled, split-face study involving 22 patients found that those who received botulinum toxin combined with broadband light on one side of the cheek showed significant reductions in the global flushing symptom score, VISIA red value, erythema index, TEWL, and sebum secretion, as well as an increase in skin hydration compared to the control group receiving broadband light plus saline and baseline. At 6 months after treatment, only sebum secretion levels returned to baseline, while the other indicators remained stable compared to 3 months after treatment [231]. Two small-sample single-arm studies also reported significant improvements in erythema and flushing, although symptom rebound occurred at 6 months, albeit not to the baseline level [232,233]. Mild and self-resolving adverse reactions were observed in all studies. Overall, while the use of botulinum toxin as a treatment for rosacea shows promise, further research with larger sample sizes and longer follow-up periods is needed to confirm its efficacy and safety.

Paroxetine is a selective 5-HT reuptake inhibitor commonly used as a psychotropic drug for treating depression [234]. However, recent studies have indicated that it may have a regulatory effect on autonomic nervous system-mediated vascular dilation and constriction [235]. In this multicenter, randomized, double-blind, placebo-controlled clinical trial, 112 patients with refractory erythema of rosacea were recruited and randomized to receive either placebo or 25 mg/day of paroxetine for 12 weeks. Among the 97 patients who completed the study, the group receiving paroxetine exhibited noteworthy amelioration in rosacea manifestations, such as erythema, flushing, and burning sensation; however, no significant improvement was observed in inflammatory lesions. The safety profile of paroxetine was consistent with previous studies, with dizziness, lethargy, nausea, dyspepsia, and muscle tremors being the most commonly reported adverse events [236]. Given that rosacea can cause significant psychological distress to patients, paroxetine may represent a promising treatment option for those experiencing concurrent symptoms, such as anxiety, depression, and insomnia.

A recent double-blind, randomized, placebo-controlled, cross-over trial has provided evidence that PACAP38 can cause prolonged facial flushing and swelling in patients with rosacea. Furthermore, the study demonstrated that sumatriptan can alleviate these features [237]. Sumatriptan is a 5-HT1B/1D receptor agonist and is widely used to treat migraines by inhibiting the degranulation of mast cells and reducing PACAP levels [237,238]. However, whether sumatriptan can be used for the clinical treatment of rosacea requires further investigation. The authors of the study also reported a case of a patient with severe and painful flushing who was successfully treated with oral sumatriptan (50 mg). The patient experienced a significant reduction in burning sensation, swelling, redness, and pain after 30–60 min of administration, and the effects persisted for several days [238].

Previous studies and case reports on the oral administration of other β-adrenergic receptor blockers, such as propranolol and carvedilol, have also shown good efficacy against ETR, particularly when associated with anxiety [239,240]. Nonselective β-blockers have been shown to reduce sympathetic activity and alleviate symptoms of anxiety in healthy individuals [241,242,243]. Carvedilol, which has both α1 receptor blocking and non-selective β receptor blocking effects, can slow the heart rate by acting on cardiac β1-adrenergic receptors, thereby reducing patient tension and anxiety [244,245]. Additionally, it exerts anti-inflammatory effects by inhibiting NLRP3 inflammasome and the expression of TLR2 in macrophages [50,246]. However, their systemic side effects, such as hypotension and bradycardia, should be taken into consideration.

### 3.4. Antimicrobial Strategies

Metronidazole has received FDA approval in various topical formulations owing to its anti-demodex and anti-inflammatory effects. Its mechanism of acaricidal action is yet to be fully understood despite its ability to reduce the density of *Demodex* mites in hair follicles. It is believed that the drug’s active metabolites formed in vivo are responsible for its therapeutic action rather than a direct effect on the mites [72,247]. In terms of its anti-inflammatory properties, the mechanism of action of metronidazole involves a reduction in neutrophil-derived ROS production, as well as acting as a scavenger of these reactive species [248,249]. Moreover, metronidazole has been shown to impair the induction of IL-17 both directly and indirectly via the suppression of IL-6 and CXCL-8 [72]. Additionally, 1% ivermectin cream has been approved by regulatory agencies in the US and Europe for treating both PPR and OR [5,6,7,197]. Its efficacy is superior to topical metronidazole, and it is more tolerable than azelaic acid [248,250]. Its primary mode of action is attributed to its ability to eliminate *Demodex* mites. Ivermectin cream can also exert anti-inflammatory effects by stimulating the production of anti-inflammatory cytokines, such as IL-10, and inhibiting pro-inflammatory cytokines, like IL-1b and TNF-α [248,251]. Although well-tolerated, the sudden death of *Demodex* mites may trigger a transient exacerbation of symptoms, which can be alleviated by short-term corticosteroid use. Oral ivermectin has also demonstrated efficacy in some cases of rosacea [2], complementing topical treatment options. Other effective topical agents, such as sodium sulfacetamide and benzoyl peroxide, are also widely utilized [2,5].

Omiganan is a synthetic antimicrobial peptide with rapidly bactericidal and fungicidal properties [252], rendering it a promising therapeutic agent for the treatment of various skin infections. A Phase III clinical trial was conducted to evaluate the efficacy and safety of omiganan gel in patients with severe PPR. In this randomized, double-blind, vehicle-controlled, parallel-group, multicenter study, patients who received omiganan gel exhibited a significant decrease in the mean inflammatory lesion counts and lower IGA scores, indicating significant improvement in the severity of their PPR. Importantly, omiganan gel was well-tolerated, and no serious adverse events were reported [253].

Rifaximin, an orally administered antibiotic with gut-specific activity and no systemic absorption, is registered for the treatment of traveler’s diarrhea and other conditions. As mentioned earlier, evidence suggests a link between rosacea and gastrointestinal disorders. Clinical trials have also indicated that about half of all rosacea patients have SIBO. Treatment with rifaximin has shown variable degrees of improvement in rosacea characteristics in SIBO patients, with different studies reporting improvement rates of 95.7%, 64.5%, and 82% [113,254,255]. Another prospective study involving 180 participants yielded different results, with a significantly higher rate of *Helicobacter pylori* infection among rosacea patients (48.9% vs. 26.7%) compared to non-rosacea patients, while the prevalence of SIBO was comparable between the two groups (10% vs. 7.8%). However, treatment with clarithromycin-containing sequential therapy to eradicate H. pylori and with rifaximin to treat SIBO led to significant improvement in skin lesions in rosacea patients, with improvement rates of 97.2% and 85.7%, respectively [114]. In some cases, rifaximin has also been reported to be effective in treating PPR or in improving rosacea during the treatment of gastrointestinal disorders [238,256].

### 3.5. Physical Therapy

Intense pulsed light (IPL), neodymium: yttrium–aluminum–garnet laser (Nd:YAG), pulsed dye laser (PDL), and potassium titanyl phosphate laser (KTP) are currently widely used and have shown effectiveness in the treatment of telangiectasia and erythema associated with rosacea. These treatment modalities primarily target sebaceous glands, hemoglobin, and pigmentation. In addition, ablative laser resurfacing techniques, including the use of CO2 or Er:YAG modalities, and surgical procedures, including electrosurgery, may be employed for the management of phymatous features of rosacea [2,5,7,197,248]. However, as different wavelengths of light target distinct objectives and produce diverse effects, there is potential for synergistic benefits when combining multiple laser therapies. A recent single-arm trial involving 68 patients confirmed the efficacy of the sequential use of a 532/1064 nm Nd:YAG laser followed by IPL in effectively managing facial telangiectasias and erythrosis [257]. Additionally, a split-face trial conducted in China observed that the sequential use of Nd:YAG laser after IPL treatment for facial telangiectasia exhibited higher clinical efficacy compared to separate therapies [258]. These findings highlight the potential advantages of tailored laser combinations for optimizing treatment outcomes in patients with rosacea.

Photodynamic therapy (PDT) is an effective approach in dermatology, utilizing specific wavelengths of light to activate photosensitizers and generate singlet oxygen and other ROS. While PDT is well-established for treating conditions like acne and Bowen’s disease, its potential for managing rosacea is currently being investigated [259]. Proposed mechanisms of action in rosacea include immune modulation, the regulation of pilosebaceous units, and targeting *Demodex* mites through porphyrin activation [260]. PDT also exhibits broad-spectrum antimicrobial effects [261,262]. Although the exact mechanisms underlying PDT’s therapeutic effects in rosacea require further investigation, a systematic review of nine studies indicates satisfactory treatment outcomes despite variations in the photosensitizers, light sources, and parameters used. Adverse reactions, although generally tolerable and temporary, were reported in some studies [263]. PDT may have superior efficacy in managing PPR compared to ETR [263,264,265]. In our latest study, we discovered that PDT exhibited similar efficacy as that of first-line oral antibiotics in the treatment of PPR, and it demonstrated stronger but non-significantly different acaricidal activity than oral antibiotics. However, the most prominent post-treatment response was erythema [266]. This observed occurrence may be explained by the heightened sensitivity of blood vessels to the thermal and radical effects induced by PDT, along with the immune response triggered by *Demodex* residues [251,267]. However, PDT-induced erythema can self-resolve within 1–2 weeks and is more controllable, with significant improvement achieved through the use of red light and IPL. Future refinements of parameters hold promise for better outcomes. Recent randomized controlled trials with larger sample sizes have reported improved treatment efficacy and milder adverse reactions when PDT is combined with other modalities, such as 1550 nm fractional therapy laser or Danzhi Xiaoyao Powder [268,269]. In a single-arm trial involving 10 patients with ETR or PPR, PDT in combination with IPL also demonstrated favorable treatment outcomes and high patient satisfaction [270]. However, further research is needed to understand the treatment mechanisms, optimize the protocols, and determine long-term safety and efficacy. Comparing the recurrence rates between PDT and first-line drugs is particularly important, considering PDT’s potential in reducing recurrence rates in other disorders involving pilosebaceous units [271].

Several other non-invasive therapies have shown promise in treating rosacea, including laser therapy, radiofrequency therapy, and ultrasound. The 577 nm pro-yellow laser, which is preferentially absorbed by hemoglobin, has exhibited notable efficacy in treating ETR and reducing the density of *Demodex* mites, with a low incidence of pigmentation and scarring [272,273,274,275,276]. Radiofrequency therapy has also shown positive effects in reducing burning sensations in patients with ETR [277,278,279,280]. Additionally, short-wave radiofrequency and fractional microneedling radiofrequency have exhibited potential in managing ETR symptoms [278,281,282]. Ultrasound, particularly microfocused ultrasound with visualization, has shown efficacy in improving telangiectasia and erythema associated with rosacea by inhibiting MMPs and restoring skin barrier function [283,284,285,286]. While several clinical trials have verified these methods, larger randomized controlled trials are necessary to assess their efficacy and safety.

### 3.6. Miscellaneous

Non-occlusive sunscreen, mild facial cleansers, and skincare products may repair the skin barrier, but evidence is inadequate [2,287,288,289]. Dietary supplementation with N-3 polyunsaturated fatty acids has shown anti-inflammatory effects and is a promising strategy for treating rosacea [290]. Stress management techniques, such as meditation, may reduce cortisol levels and alter the course of skin diseases like rosacea fulminans [291]. Several traditional herbal medicines, including turmeric, celastrol, diammonium glycyrrhizinate, licochalcone A, and extracts of tormentil, simarouba amara, and supramolecular salicylic acid, have broad-spectrum anti-inflammatory mechanisms in treating rosacea [292,293,294,295,296,297]. The emergence of these herbal medicines may offer a safer alternative or complement conventional pharmacotherapies to achieve optimal therapeutic outcomes [292,294]. However, further research is needed to explore their mechanisms in depth.

Small-molecule inhibitors are at the forefront of pharmacological biomedical therapeutics. Recently, Ogasawara et al. identified two MRGPRX2 antagonists that have the potential to prevent IgE-independent allergic reactions by specifically blocking the activation of human mast cells mediated by MRGPRX2 [298]. While their efficacy for treating rosacea remains to be investigated, these novel inhibitors hold promise for the development of new drugs for rosacea treatment. Alternatively, RNA medicines offer an opportunity to manage rosacea conditions at the transcriptional level with greater specificity and design flexibility than small molecules. However, the delivery of RNA medicines to the cytoplasm of target cells non-invasively in rosacea patients is a significant challenge. To address this, Colombo et al. developed a small RNA interference (siRNA) that targets TLR2 and applied emulsified siRNA to the inner and outer surface of mice ears in the presence of active excipients, such as glycerol/urea. This resulted in a significant decrease in TLR2 levels, indicating the potential for siRNA to manage rosacea at the transcriptional level [299]. These findings suggest that such drugs may represent the future direction of rosacea management.

**Table 1 biomedicines-11-02153-t001:** Therapeutic strategies targeting diverse pathogenic mechanisms. An asterisk (*) denotes well-established therapies, typically approved by the FDA or recommended by multiple national guidelines or expert consensus.

Target	Management Options	Pharmacological Effects	Current Clinical Trials
Immune Dysregulation	Azelaic acid *	Suppresses expression of KLK5 and cathelicidin, activates PPARγ to exhibit anti-inflammatory properties, and curbs expression of IL-1, IL-6, and TNF-α	FDA-approved
	ε-aminocaproic acid	Inhibits KLK5	Shows beneficial impact on the severity of rosacea in a small, randomized pilot trial [192]
	Doxycycline (sub-antibiotic doses) *	Inhibits chemotaxis and ROS production in neutrophils, suppresses several MMPs and subsequent antimicrobial peptide production, targets abnormal amino acid metabolism and sebaceous gland cells	FDA-approved
	Isotretinoin *	Modulates TLR2 expression negatively in keratinocytes, reduces sebum production and sebaceous gland size	Supported by guidelines or expert consensus [2,5,7,197]
	Pimecrolimus *	Inhibits T cell and mast cell activation by blocking calcineurin action	Supported by guidelines or expert consensus [2]
	Tacrolimus	Inhibits calcineurin	Clinical trials conducted with varying numbers of participants (1 to 200) in a systematic review of 28 articles [201]
	Hydroxychloroquine	Attenuates LL37-induced mast cell activation partly by inhibiting calcium influx	Small-sample, multicenter randomized controlled trial comparing hydroxychloroquine to standard doxycycline treatment showed similar efficacy and safety [202]
	Artemether	Suppresses expression of inflammatory biomarkers induced by LL37 via inhibition of transcription factors NF-κB, mTOR, and STAT	Randomized pilot study including 130 subjects evaluated efficacy of artemether emulsion [206]
	Tranexamic acid	Suppresses expression of KLK5, Camp, and TLR2, suppresses expression of cytokines and chemokines, inhibits angiogenesis induced by LL37, inhibits serine protease and physical interaction between urokinase-type plasminogen activator and the stratum corneum	Small-sample clinical trials and case studies have shown effectiveness of tranexamic acid, administered via different routes [209,210,211,214]
	ACU-D1	Inhibits activation of NF-κB	Shows efficacy in double-blind, randomized, placebo-controlled study involving 40 patients [215]
	Secukinumab	Blocks IL-17	Small open-label study conducted [217]
Vascular Dysregulation	Brimonidine tartrate *	α2-adrenergic receptor agonist, promotes contraction of vascular smooth muscle cells	FDA-approved
	Oxymetazoline *	α1-adrenergic receptor agonist, promotes contraction of vascular smooth muscle cells	FDA-approved
	Timolol	Nonselective β-adrenergic receptor blocker, induces vasoconstriction, inhibits inflammatory mediators such as MMPs and IL-6, inhibits angiogenesis by downregulating VEGF, promotes migration and re-epithelialization of keratinocytes, affects the secretion of lamellar bodies mediating repair of the skin barrier	Pilot trial found long-term topical use improved erythema, but rebound occurred after discontinuation; small trial showed significant improvement in erythema with nightly use for 28 days; larger trial found improvement in clinical parameters, but did not reach statistical significance [223,224]
Neurological and Psychological Factors	Botulinum toxin	Inhibits release of vasodilating acetylcholine, regulates NPs such as SP, CGRP, and VIP, reduces mast cell count and degranulation, decreases expression of certain MMPs, reduces sebum production, and increases skin hydration	Limited clinical trials with small sample sizes, imperfect study designs, and short follow-up times suggest potential efficacy and safety for rosacea treatment [230,231,232,233]
	Paroxetine	Inhibits the reuptake of 5-HT	Demonstrated efficacy in a multicenter randomized controlled trial [236]
	Sumatriptan	5-HT1B/1D receptor agonist, inhibits degranulation of mast cells, reduces PACAP levels	Alleviates features of rosacea in double-blind, randomized, placebo-controlled, cross-over trial and successful treatment of severe and painful flushing in a single case report [237,238]
	Propranolol	β-adrenergic receptor blocker, reduces sympathetic activity and alleviates anxiety symptoms	Beneficial impact in some small-sample studies and case reports [239,240]
	Carvedilol	Has both α1 receptor blocking and non-selective β receptor blocking effects, slows heart rate by acting on cardiac β1-adrenergic receptors to reduce patient tension and anxiety, and exerts anti-inflammatory effects by inhibiting NLRP3 inflammasome and the expression of TLR2 in macrophages	A large-scale randomized controlled trial showed that oral carvedilol exhibited better efficacy than topical brimonidine [244,245]
Microbial Dysbiosis	Metronidazole *	Exerts acaricidal effects via its active metabolites, reduces ROS production and scavenges reactive species, impairs IL-17 induction	FDA-approved
	Ivermectin *	Eliminates *Demodex* mites, reduces neutrophil response, stimulates production of anti-inflammatory cytokines such as IL-10, inhibits pro-inflammatory cytokines like IL-1b and TNF-α	FDA-approved
	Omiganan	Rapidly kills bacteria and fungi	Phase III clinical trial showed effectiveness and safety in severe papulopustular rosacea [253]
	Rifaximin	Treats SIBO by inhibiting bacterial RNA synthesis	Several clinical trials and case reports have shown that rifaximin effectively improves rosacea characteristics in SIBO patients [113,114,238,254,255,256]
Physical Therapy	IPL, Nd:YAG, PDL, and KTP *	Primarily targets sebaceous glands, hemoglobin, and pigmentation	Supported by guidelines or expert consensus [2,5,7,197,248]
	Ablative laser resurfacing *	Targets water, causes vaporization and ablation effects	Supported by guidelines or expert consensus [2,5,7,197,248]
	Photodynamic therapy	Activates photosensitizers with light to generate ROS, modulates immunity and pilosebaceous units, targets *Demodex* mites, and exhibits antimicrobial effects	Systematic review of nine Level 4 studies suggests PDT may be a safe and effective treatment option; findings from ongoing and smaller-scale trials indicate that PDT may offer efficacy comparable to that of first-line therapies in addressing PPR; results from larger randomized controlled trials combining PDT with other modalities indicate improved efficacy and milder adverse reactions [263,264,265,268,269,270]
	Pro-yellow laser	Emits laser with a wavelength of 577 nm, demonstrating preferential absorption by hemoglobin	Demonstrated efficacy in select case reports and small sample trials; a retrospective study identified reduction of mite density [272,273,274,275,276]
	Radiofrequency	Generates thermal energy, has positive effects on the nervous system, cardiovascular system, immune system, and reduces burning sensations by decreasing TRPV1 expression	Randomized, controlled, split-face study showed radiofrequency and PDL equally effective in treating ETR; radiofrequency treatment showed greater improvement in PPR [280]
	Short-wave radiofrequency	Enhances local blood oxygen supply, repairs skin barriers, and reduces chronic inflammation	Prospective, single-arm, open-label pilot study reported rapid and sustained improvement in mild to moderate ETR patients [278]
	Fractional microneedling radiofrequency	Delivers thermal energy through targeted microneedles, reduces dermal inflammation, mast cell count, and the expression of TLR2, LL37, VEGF, NF-κB, IL-8, and TRPVs	Prospective, randomized, split-face clinical trial showed modest but statistically significant improvement in rosacea [281]
	Ultrasound	Restores skin barrier function by inhibiting MMPs	Both retrospective and prospective studies have reported significant improvements in patient self-assessment and clinical measures [283,284,285,286]

## 4. Conclusions

In this comprehensive review, we present a thorough summary of the current knowledge and recent advancements regarding the pathogenesis and treatment of rosacea. Our exploration begins by providing detailed insights into the two most well-established aspects: (1) immune dysregulation and (2) neurovascular dysregulation. Subsequently, we conducted an in-depth analysis of (3) neuroimmune dysregulation, (4) skin barrier dysfunction, (5) local and systemic microbial dysbiosis, (6) metabolic dysfunction, and (7) sebaceous gland dysfunction. Furthermore, we have highlighted the significant contributions of (8) genetic predisposition, (9) oxidative stress, and (10) environmental triggers to the complex landscape of rosacea pathogenesis.

We have highlighted the presence of multiple positive feedback loops within immune dysregulation, leading to persistent inflammation. In addition, we have underscored the complex interconnections between the major pathogenic mechanisms underlying rosacea, creating a vicious cycle that promotes the development of rosacea. For example, changes in skin microbiota and resulting inflammation can lead to subsequent alterations in sebaceous gland function and skin barrier disruption, further modifying microbiota composition. As such, we emphasize the crucial role of sebaceous gland dysfunction in these interactions. Additionally, we draw attention to the noteworthy interplay between the nervous and immune systems, particularly in the context of refractory and recurrent rosacea, which can present with neurological symptoms and even psychiatric disorders such as depression and anxiety. These issues not only significantly affect the quality of life of patients but also have the potential to exacerbate the disease. To address these challenges, we require animal models that accurately simulate the regulatory role of the nervous system in rosacea. Thus, this direction is a key area of focus for future research. Furthermore, studies on metabolism, genetics, psychological, and other factors indicate that rosacea should not be considered solely as a facial disorder. Given the multifaceted pathogenic mechanisms involved in rosacea, investigating these emerging areas may offer novel therapeutic avenues for the condition. Our future goal is to pinpoint the key molecules or mechanisms that drive inflammation in rosacea, akin to the role of IL-17 in psoriasis, and to develop therapeutic agents based on these findings.

Recent advances in our understanding of the pathogenesis of rosacea have led to the emergence of various new therapies. In this paper, we have dedicated comprehensive chapters to elaborate on the current understanding and recent advancements in therapeutic strategies that specifically address (1) immune dysregulation, (2) neurovascular dysregulation, (3) neurological and psychological factors, and (4) microbial dysbiosis. Additionally, we have meticulously explored the latest developments in (5) physical treatment methods, encompassing photodynamic therapy and other innovative approaches. Moreover, we have delved into (6) miscellaneous therapeutic avenues, including the promising utilization of traditional herbal medicines, small-molecule inhibitors, and RNA medicines. These promising therapies have enriched the range of available treatment options, providing new avenues for managing the complex pathophysiology of rosacea. Nonetheless, the efficacy of many of these novel therapies necessitates further validation through rigorous clinical trials. Some physical therapies have also emerged as potential avenues for future development. These therapies target specific symptoms with minimal systemic adverse effects, making them suitable for combination with other therapeutic modalities or post-pharmacological intervention. Tailored combinations of physical therapies present advantages in optimizing treatment regimens for rosacea patients and may contribute to improved aesthetic outcomes. Given that rosacea may extend beyond the skin, personalized therapies that target the comorbidities associated with rosacea, such as β-adrenergic receptor antagonists for patients with anxiety and rifaximin for those with SIBO, could be another promising direction for the future of rosacea treatment. By addressing the individual needs of patients with specific comorbidities, these therapies have the potential to provide more effective and tailored treatment options. With the emergence of monoclonal antibodies, small-molecule drugs, and RNA medicines, we now have more precise drugs that target the disease development process. Therefore, exploring the core molecules involved in the pathogenesis of rosacea may lead to the development of revolutionary drugs that address the root cause of the disease. This paradigm may serve as the primary avenue for advancing future treatment strategies for rosacea.

## Figures and Tables

**Figure 1 biomedicines-11-02153-f001:**
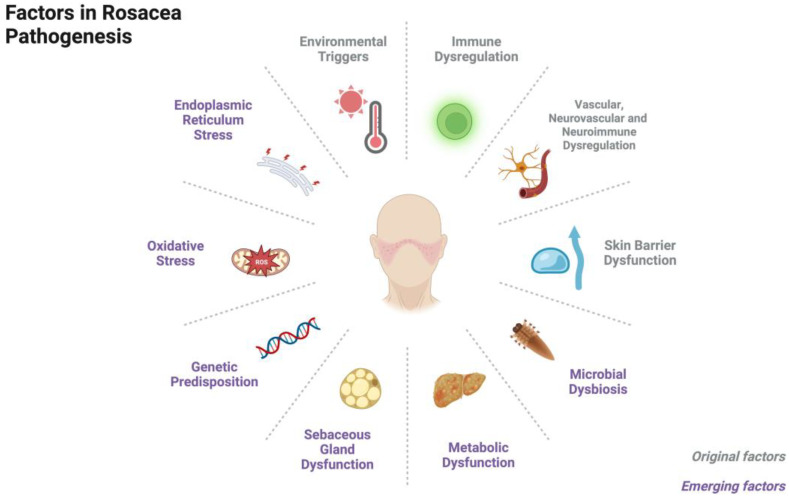
Schematic diagram illustrating the original and emerging factors implicated in the pathogenesis of rosacea. The figure was created with BioRender.com (accessed on 28 June 2023).

**Figure 2 biomedicines-11-02153-f002:**
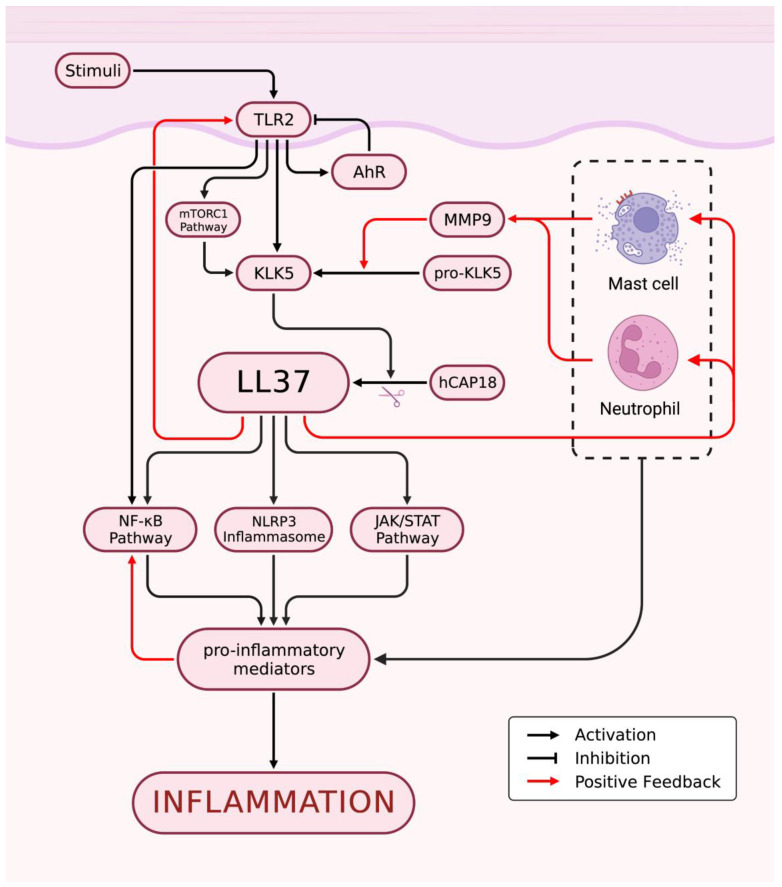
Schematic representation of TLR2/KLK5/LL37 pathway. The red arrows in the figure signify a positive feedback loop, which contributes to the chronicity of the disease. The figure was created with BioRender.com (accessed on 28 June 2023).

**Figure 3 biomedicines-11-02153-f003:**
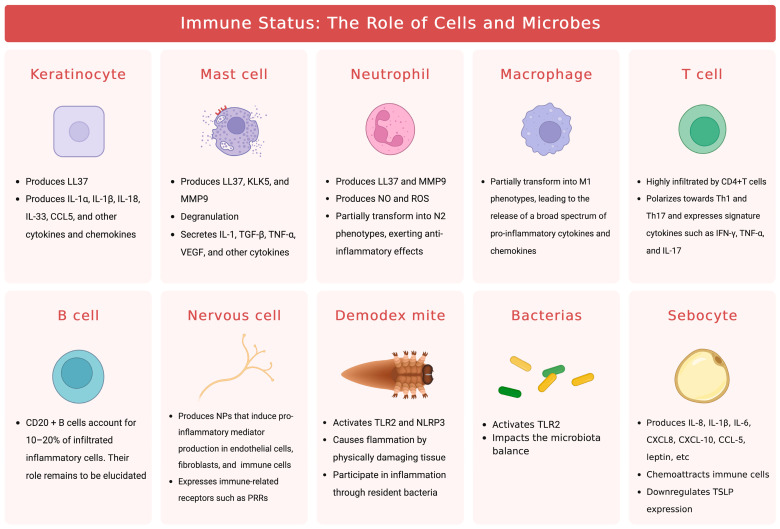
Cells and microbes involved in the immune state of rosacea. The figure was created with BioRender.com (accessed on 28 June 2023).

**Figure 4 biomedicines-11-02153-f004:**
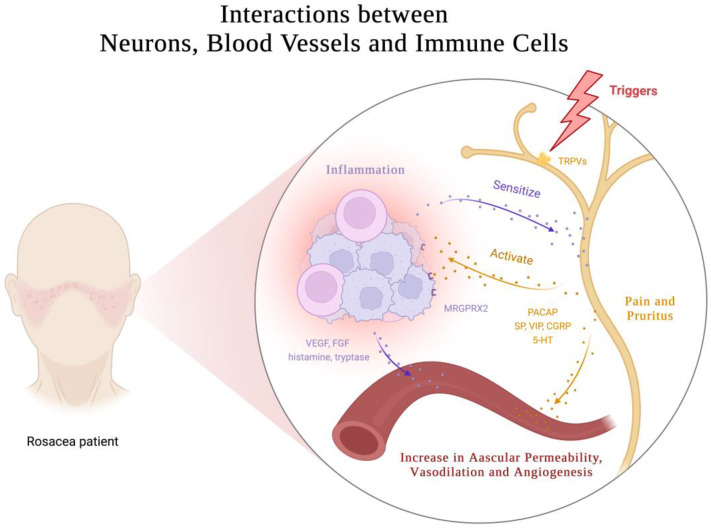
Schematic illustration depicting the intricate interplay among nerves, blood vessels, and immune cells in the development of rosacea. The figure was created with BioRender.com (accessed on 28 June 2023).

**Figure 5 biomedicines-11-02153-f005:**
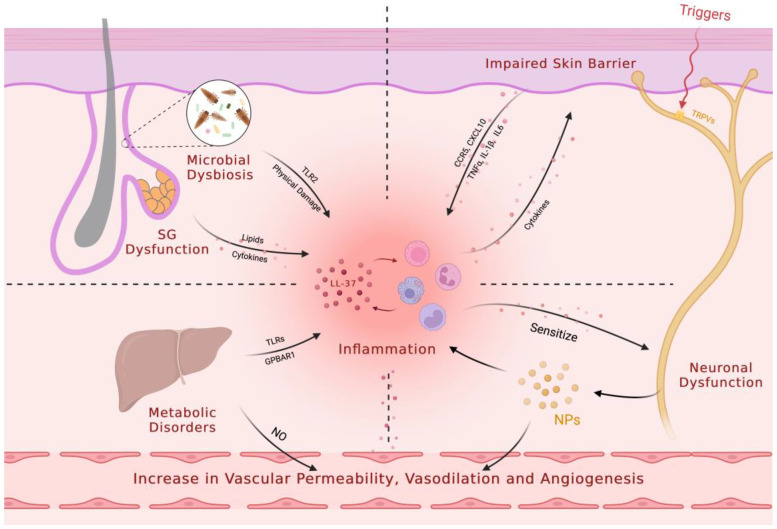
Major factors involved in the pathophysiology of rosacea. The figure was created with BioRender.com (accessed on 28 June 2023).

## Data Availability

This research did not involve the collection, use, or access of any data. Therefore, no data access statement is required.

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
