# Peer review of "Exploring the Pathogenesis and Mechanism-Targeted Treatments of Rosacea: Previous Understanding and Updates"

_biomedicines, 2023, doi:10.3390/biomedicines11082153_

Round 1

Reviewer 1 Report

The manuscript “Exploring the Pathogenesis and Mechanism-Targeted Treatments of Rosacea: Previous Understanding and Updates” by  Chengqian Chen et al is an interesting review focused attention on  the pathogenesis of rosacea and particularly on emerging and novel mechanisms. Finally it provided an up-to-date overview of treaments.

               The review is interesting and in-depth especially on the pathogenetic mechanisms, there are beautiful explanatory images. As far as therapy is concerned, the authors focused above all on emerging therapies which they also summarized in an explanatory table.

               I think it would be useful to further develop the section dedicated to “well-established therapies”, emphasizing those which have proved to be more useful and effective over time, again summarizing them in a summary table to have a more complete overview of the disease

Author Response

Dear Reviewer,

Thank you for your insightful comments and valuable feedback on our manuscript titled "Exploring the Pathogenesis and Mechanism-Targeted Treatments of Rosacea: Previous Understanding and Updates." We appreciate your positive evaluation of the review and are grateful for the opportunity to enhance the content to provide a more comprehensive overview of rosacea treatments.

We understand the significance of emphasizing "well-established therapies" in the context of treating rosacea. Based on your suggestion, we have revised the sentences related to these therapies to highlight their proven efficacy and usefulness over time. In the respective sections, we have added phrases such as "a recommended option," "approved by the FDA," "widely used," and "in accordance with Swiss guidelines" to underscore their status as well-established therapies. Additionally, for drugs briefly mentioned earlier in the manuscript, such as isotretinoin, metronidazole, and ivermectin, we have provided more extensive information to emphasize their clinical importance.

However, rather than consolidating these well-established therapies into a separate section, we have opted to integrate them throughout the corresponding sections, with a focus on their relevance to specific pathogenic mechanisms. We believe that organizing treatments based on different mechanisms aids in maintaining the clarity of the manuscript's structure and enhances the connection between various sections, which may facilitate future drug development efforts. We appreciate your valuable suggestion and apologize for any confusion caused by this decision.

Furthermore, in the summary table, we have marked these well-established therapies with an asterisk (*) to enable readers to quickly differentiate them from emerging therapies.

Once again, we express our gratitude for your thoughtful review, which has undoubtedly strengthened our manuscript. We hope that our revised version now offers a more thorough and informative overview of rosacea pathogenesis and treatments.

Sincerely,

Chengqian Chen

Institute of Photomedicine, Shanghai Skin Disease Hospital, School of Medicine, Tongji University, Shanghai 200040, China

Reviewer 2 Report

Review of manuscript ID biomedicines-2507366 and titled “Exploring the Pathogenesis and Mechanism-Targeted Treatments of Rosacea: Previous Understanding and Updates”.

In my opinion, the work fits into the framework of the journal and deals with an important issue. It is well written and also very well designed graphically. As a reviewer, I sometimes evaluate works that are well written, but their graphic part is at a poor level. In this situation, both the substantive and graphic part are at a high level. In connection with the above, I recommend the works for acceptance and congratulate the authors on a job well done.

I noticed some minor flaws, but they do not affect the reception of the work.

Author Response

Dear Reviewer,

Thank you for your positive feedback and recommendation for acceptance. We are glad you found our work fitting for the journal and appreciate your recognition of both the content and graphical design. Your encouragement motivates us to continue contributing to the field.

Sincerely,

Chengqian Chen

Institute of Photomedicine, Shanghai Skin Disease Hospital, School of Medicine, Tongji University, Shanghai 200040, China

Reviewer 3 Report

 Very intersting and practical review.  Congratulations!

1. In this review, Authors present a thorough summary of current knowledge and recent advancements regarding the pathogenesis and treatment of rosacea. 2. Authors highlight the presence of multiple positive feedback loops within the immune dysregulation, leading to persistent inflammation. In addition, They underscore the complex interconnections between the major pathogenic mechanisms underlying rosacea, creating a vicious cycle that promotes the development of rosacea. 3. Authors draw attention to the noteworthy interplay between the nervous and immune systems, particularly in the context of refractory and recurrent rosacea, which can present with neurological symptoms and even psychiatric disorders such as depression and anxiety. These issues not only significantly affect the quality of life of patients but also have the potential to exacerbate the disease. To address these challenges, they require animal models that accurately simulate the regulatory role of the nervous system in rosacea. Thus, this direction is a key area of focus for future research. 4. Not applicable 5. Not applicable 6. The reference (292 positions) are appropriate. 7. No comments

Author Response

Dear Reviewer,

Thank you for taking the time to review our manuscript titled "Exploring the Pathogenesis and Mechanism-Targeted Treatments of Rosacea: Previous Understanding and Updates." We are delighted to receive your positive feedback and appreciate your acknowledgment of the overall significance and practicality of our work.

We are grateful for your thorough understanding and grasp of the main ideas and key points presented in the manuscript. Your encouraging remarks serve as motivation for our continued efforts in contributing valuable research to the field.

Once again, we express our gratitude for your time and insightful review.

Sincerely,

Chengqian Chen

Institute of Photomedicine, Shanghai Skin Disease Hospital, School of Medicine, Tongji University, Shanghai 200040, China

Reviewer 4 Report

Dear authors

Happy day

The paper need some more work to be in the final accepted form

1- All species scientific names must be italic.

2- Avoid as you could the over-estimated words such as ''many'' ''highest'' etc.

3- Be more specific and not generalize 

e.g. : Rosacea is a common chronic inflammatory cutaneous disorder that affects about 5.46% of the population'' Which populations?

4- Adjust the images resolutions. 

5- Summarize in points the casus and the treatments and the protection if found.

6- kindly give more size to the most important point the ''Genetic Predisposition''.

7- Give all points the same space.

 with my pleasure

Author Response

Dear Reviewer,

We hope this message finds you well. We sincerely appreciate your feedback and constructive comments on our manuscript titled "Exploring the Pathogenesis and Mechanism-Targeted Treatments of Rosacea: Previous Understanding and Updates."  We have carefully addressed each of your suggestions, and we are pleased to present the revisions made to enhance the quality of our paper.

All species scientific names must be italic.

Response 1: We have made the necessary adjustments, and all species scientific names, including Demodex, Demodex folliculorum, Demodex brevis, Bacillus oleronius, Staphylococcus epidermidis, and Helicobacter pylori, have been italicized throughout the manuscript.

Avoid over-estimated words such as ''many'' ''highest'' etc.

Response 2: We have diligently avoided using over-estimated words such as "many," "highest," "best," "all," "always," and "absolutely." Instead, we have used more cautious language, employing words such as "may" and "could."

Be more specific and not generalize

e.g.: Rosacea is a common chronic inflammatory cutaneous disorder that affects about 5.46% of the population'' Which populations?

Response 3: We appreciate your keen observation. To clarify, the 5.46% prevalence refers to the global adult population. The data was obtained from a systematic review, encompassing 32 studies with a total of 41 populations and 26,519,836 individuals.

Adjust the images resolutions.

Response 4: All images have been adjusted to have a resolution of 600dpi, ensuring optimal clarity and visual appeal.

Summarize in points the cases and the treatments and the protection if found.

Response 5: To enhance readability, we have summarized the pathogenesis and treatments in points in the Conclusions section.

Kindly give more size to the most important point, "Genetic Predisposition."

Response 6: We have expanded the "Genetic Predisposition" section, incorporating relevant insights from a recent seminal study that establishes a novel link between genetic underpinnings and neural mechanisms of rosacea. This inclusion further reinforces the significance of genetic factors in the development of the condition.

Give all points the same space.

Response 7: While we have made efforts to maintain consistency in the length of each section, it is challenging due to the extensive research available on the inflammatory aspects of Rosacea, making it the most well-studied and classical area. On the other hand, emerging factors such as "Skin Barrier Dysfunction" currently have relatively limited research available. Nevertheless, we have made necessary adjustments, and in sections 3.4 "Antimicrobial Strategies" and 3.5 "Physical Therapy," we have added relevant content to balance the word count across sections.

We sincerely thank you for your time and invaluable suggestions, which have significantly improved the manuscript. We believe that the revised version is now well-structured and more informative, contributing to the existing knowledge on rosacea.

With warm regards,

Chengqian Chen

Institute of Photomedicine, Shanghai Skin Disease Hospital, School of Medicine, Tongji University, Shanghai 200040, China

Reviewer 5 Report

A very interesting narrative review regarding pathogenesis and possible treatments of rosacea. Only minor revisions: I would implement the conclusions also talking about future perspectives and development in the management of rosacea.

I would also improve the physical treatment part, talking about combination treatments; here is an interesting article you should cite: doi: 10.3390/medicina58050651.

Thank You

Author Response

Dear Reviewer,

We sincerely appreciate your valuable feedback on our review titled "Exploring the Pathogenesis and Mechanism-Targeted Treatments of Rosacea: Previous Understanding and Updates." We are grateful for your positive remarks and have carefully considered your suggestions for improvement. Following your recommendations, we have made the necessary revisions to enhance the manuscript. Below, we provide a detailed response to each of your comments:

Implement the conclusions also talking about future perspectives and development in the management of rosacea.

Response 1: Thank you for this valuable suggestion. We have now expanded the conclusions section to include discussions on future perspectives and developments in the management of rosacea. Specifically, we have highlighted the potential of physical therapies, personalized treatments targeting comorbidities, and the use of targeted therapies focusing on core molecules as promising areas for future research and advancements in rosacea management. By incorporating these future perspectives, we aim to provide a comprehensive outlook for researchers and clinicians, highlighting potential directions for further investigation and therapeutic strategies in the field of rosacea management.

Improve the physical treatment part, talking about combination treatments; here is an interesting article you should cite: doi: 10.3390/medicina58050651.

Response 2: We are grateful for bringing this article to our attention. It has indeed provided valuable insights into combination treatments for rosacea. To address this suggestion, we have enriched the "3.5 Physical Therapy" section by incorporating relevant information on combination treatments and cited the article you recommended. This addition allows us to present a more comprehensive review of physical therapies, encompassing the potential benefits of combining various treatment modalities for optimal outcomes in rosacea management.

Once again, we sincerely appreciate your time and effort in reviewing our manuscript. Your feedback has significantly enriched the content and scope of our review, making it more comprehensive and valuable to the readers. We hope that the revised version meets your expectations and provides a more in-depth exploration of the pathogenesis and treatments of rosacea.

Sincerely,

Chengqian Chen

Institute of Photomedicine, Shanghai Skin Disease Hospital, School of Medicine, Tongji University, Shanghai 200040, China